# Modelling the Thermal Effects on Structural Components of Composite Slabs Under Fire Conditions

**Carlos Balsa** [1,*,†] **, Matheus Silveira** [2,†] **, Valerian Mange** [3,†] **and Paulo A. G. Piloto** [4,†]

1    Research Centre in Digitalization and Intelligent Robotics (CeDRI), Instituto Politécnico de Bragança, 5300-253 Bragança, Portugal
2    Campus Pato Branco, Universidade Tecnológica Federal do Paraná, Via do Conhecimento, s/n-KM 01-Fraron, Pato Branco 85503-390, Brazil; matheussilveira@alunos.utfpr.edu.br
3    Ecole Nationale Supérieure d'Électrotechnique, d'Électronique, d'Informatique, d'Hydraulique et des Télécommunications, Institut national polytechnique de Toulouse, Université de Toulouse, CEDEX 7, 31071 Toulouse, France; valerian.mange@etu.toulouse-inp.fr
4    Laboratório Associado de Energia, Transportes e Aeronáutica, Instituto Politécnico de Bragança, 5300-253 Bragança, Portugal; ppiloto@ipb.pt
*    Correspondence: balsa@ipb.pt
†    These authors contributed equally to this work.

**Abstract:** This paper presents a finite-element-based computational model to evaluate the thermal behaviour of composite slabs with a steel deck submitted to standard fire exposure. This computational model is used to estimate the temperatures in the slab components that contribute to the fire resistance according to the load-bearing criterion defined in the standards. The numerical results are validated with experimental results, and a parametric study of the effect of the thickness of the concrete on the temperatures of the slab components is presented. Composite slabs with normal or lightweight concrete and different steel deck geometries (trapezoidal and re-entrant) were considered in the simulations. In addition, the numerical temperatures are compared with those obtained using the simplified method provided by the standards. The results of the simulations show that the temperatures predicted by the simplified method led, in most cases, to an unsafe design of the composite slab. Based on the numerical results, a new analytical method, alternative to the simplified method, is defined in order to accurately determine the temperatures at the slab components and, thus, the bending resistance of the composite slabs under fire conditions.

**Keywords:** energy equation; finite element method; composite slab; standard fire; fire rating

## 1. Introduction

Steel–concrete composite slabs are made of a profiled steel deck, which can be used as a permanent formwork, and reinforced concrete. Normally, the concrete is reinforced with an anti-crack mesh positioned on the upper part and individual reinforcement bars in the ribs (see Figure 1). These construction elements offer some advantages for the structures, such as reducing the dead weight while speeding up the construction process. The use of composite slabs in buildings has become very popular in North America since 1960. However, as there was insufficient information on structural safety in Europe, this element type has became popular after 1980. The overall thickness of the slab usually varies between 100 and 170 mm. The thickness and geometry of the steel deck depends on the manufacturer of the steel deck profiles and usually includes a zinc layer on the exposed surface to increase the corrosion resistance [1].

Besides being exposed to a corrosive environment, the composite slabs may suffer considerable damage when exposed to fire, as the steel elements responsible for the bending strength of the composite slabs are significantly affected by elevated temperatures. It is therefore necessary to carry out a thermal analysis prior to the static analysis to ensure that

this building element is fire resistant in accordance with the regulations and standards. The fire resistance of composite slabs with a steel deck is usually determined by standard fire tests, considering the ability to sustain the load (R), the thermal insulation (I), and the integrity (E). In order to demonstrate the fire resistance according to the fire ratings of the European standard EN 13501-2 [2], the element must be able to prevent large deformations or rates of deformation for the case of load-bearing (R) and also provide thermal insulation that limits the temperature rise on the unexposed side (I). Finally, the composite slabs must contain the fire from below and prevent the passage of flames and hot gases through cracks or holes, keeping the integrity criterion (E). This last criterion is usually verified by considering the presence of the steel deck.

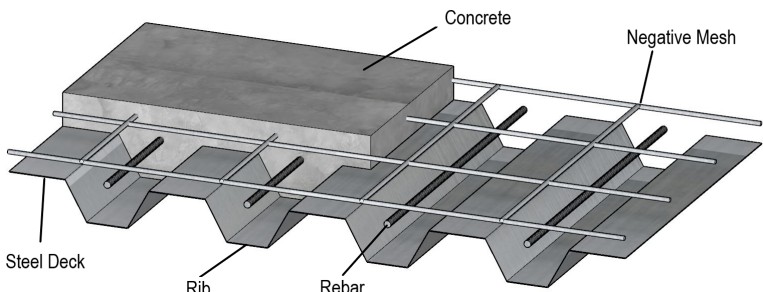

**Figure 1.** Composite slab with trapezoidal steel deck.

This paper is concerned with the calculation of the temperature of composite slabs under a standard, ISO-834 fire [3], in particular the temperature evolution at the components of the steel deck (upper flange, web, lower flange) and the reinforcing bars. An accurate and reliable method to determine the temperature field in these components is required, in particular to determine the load-bearing criterion (R), as these temperatures have a direct influence on the reduction factors for the steel and concrete strength and, consequently, on the bending resistance of the slabs.

Among the different methods to determine the fire resistance of a composite slab, the development of standard experimental fire tests is the most expensive and time consuming. Alternatively, Annex D of EN 1994-1-2 [4] provides the guidelines to determine the fire resistance based on the simplified calculation method. This method is based on studies conducted a long time ago and is currently outdated. The third method is to simulate the experimental fire tests computationally using numerical methods. Computer simulations are of great importance in this field because they allow a reliable and realistic description of the physical phenomena, including the effects of different fire scenarios, such as natural fires. In addition, computer simulations are less expensive and faster than conducting experimental tests.

The year 1983 was very important for the spread of composite slabs in Europe as an alternative and functional building method. In that year, the European Convention for Constructional Steelwork ECCS [5] published the design rules for composite concrete slabs with a steel deck exposed to a standard fire. According to this document, explicit fire design is not required for composite slabs when the fire requirements are less than or equal to 30 min, and for other fire ratings, the calculation methods are based on conservative approximations for a safer design procedure. This technical guide has been developed to provide structural safety in accordance with the fire resistance classes specified in ISO 834-1 [3] without the need for experimental testing. This document should only be used if the composite slab has been safely designed for room temperature.

Hamerlinck developed some mathematical models for estimating the fire resistance of composite slabs. In 1990, he proposed a numerical procedure comprising thermal models for cross-section analysis and a structural analysis model for composite slabs. In his paper published in 1991 [1], Hamerlinck described in detail all the methods for carrying out numerical and experimental studies regarding the thermal and mechanical behaviour of reinforced composite slabs under fire. The experimental programme took into account

the most important parameters for fire resistance, and a new computer programme was developed to allow simulations with low computational cost (low processing time). It was concluded that the two-dimensional model developed gave satisfactory results, although it did not take into account the three-dimensional thermal effects. The current standards are still based on his work.

In 1998, Both [6] introduced an easy-to-use calculation rule in order to give more insight into the fire behaviour and failure mechanisms of continuous composite slabs. The numerical models were validated against the results of experimental tests conducted by the author and other researchers. The numerical models were created using the multi-purpose finite element programme DIANA, originally developed at TNO Building and Construction Research. The 2D and 3D thermal models of the composite slab were developed. The authors concluded that the thermal model was capable of describing the two- and three-dimensional heat flow in composite slabs during fire exposure and that the assessment rules for fire resistance, which at that time were specified in Eurocode 4, could be significantly improved.

In recent years, a wide range of thermal and structural models have been developed in various finite element software programmes to predict the structural and thermal behaviour of steel–concrete composite slabs under fire conditions. In 2018, Piloto et al. [7] analysed the fire resistance of composite concrete slabs with a profiled steel deck, in this case also with a steel mesh on the top, including reinforcing bars between the ribs. The main objective of this study was to develop two-dimensional numerical models using the Matlab and ANSYS programmes to evaluate the fire resistance of different composite slab configurations according to the insulation criterion. For the development of this research, the fire resistance criterion for insulation (I) was evaluated using numerical and simple calculation methods and then validated using experimental fire tests. The results obtained by the authors allowed proving that the fire resistance (I) increases with the concrete thickness for both calculation methods. However, when using the numerical method, the simulation predicts a lower fire resistance (I) compared to the basic standards, thus ensuring a higher structural safety with this method. Therefore, a new and better approach has been proposed that considers a quadratic variation between the fire resistance and the effective thickness of the composite slab. For the first time, the authors [8] decided to consider an air gap between the steel deck and the concrete topping. This innovation considers the implicit model of the detachment between the steel deck and the concrete, which has been demonstrated in many experimental fire tests.

In 2019, Jian Jiang et al. [9] from the National Institute of Standards and Technology (NIST) conducted a numerical investigation on various parameters that may affect the fire resistance of composite slabs with respect to the thermal insulation criterion (I). A set of 54 composite slabs was selected for the numerical analysis, which were developed using the finite element software LS-DYNA. A 2D high-fidelity thermal model of the symmetrical part of the cross-section of the composite slab was used. The concrete thickness and moisture content were found to be the parameters that most influenced the fire resistance. An improved algebraic expression for the calculation of the fire resistance was proposed that explicitly takes into account the moisture content of the concrete. The formulation is applicable to an extended range of geometries compared to the limitations of the calculation method in the current version of Eurocode 4.

Recently, full-scale 3D numerical models have been developed by Paulo Piloto and his team to evaluate the thermal behaviour of a series of composite slabs with different geometries under the action of different types of standard fires. In [10–12], 3D finite element models were developed to perform thermal analyses based on the insulation criterion (I), and in [11,13], a coupled thermal and mechanical model was applied to perform mechanical analyses based on the load-bearing criterion (R). The thermal models were implemented using Matlab or ANSYS software, and the mechanical model was developed using ANSYS only. In both cases, three-dimensional finite elements were used.

In 2021, Bolina et al. [14] determined the temperature analysis of steel–concrete slabs exposed to a standard fire. The main objective of this investigation was to develop a comparison of the temperature evolution in the cross-section through different methods: experimental, numerical, and simplified. The authors conducted eight full-scale fire tests, which were used to validate the numerical models developed with the ABAQUS software. The numerical results were compared with the analytical results obtained from Eurocode 1994-1-2 [4]. The authors found a good agreement between the simplified method and the other methods (numerical and experimental) only for the steel deck temperature, but not for the concrete, the reinforcing bars (positive and negative), and the thermal insulation. Therefore, a new approximation was developed to determine the temperature in the rebars, including new factors to evaluate the performance of the thermal insulation.

In the present paper, which is an improved version of the conference paper [15], the authors improve the analytical temperature calculation method, following the simplified calculation method in Annex D of Eurocode 1994 1-2. This analytical method allows the calculation of temperatures in different composite slab components (lower and upper flange, web and reinforcing bars) for different fire ratings. A correct estimation of these temperatures is of great importance for the design of the slabs according to the load-bearing criterion. Since the temperatures affect the mechanical properties of the slabs, incorrect temperature values can lead to unsafe design. A parametric study is carried out to analyse the thermal behaviour of composite slabs as a function of concrete thickness. The temperature distribution along the composite slabs of normal and lightweight concrete (NWC and LWC) was determined using four different types of composite slabs selected to represent two different steel deck geometries, trapezoidal and re-entrant. The composite slabs were subjected to the thermal effects of a standard fire exposure from below.

The full-scale tests were simulated with three-dimensional finite elements using the Matlab Partial Differential Equations Toolbox (PDE Toolbox) [16]. The discrete composite slab models consist of different physical subdomains with different thermal properties corresponding to the components of the slab such as concrete, steel deck, and reinforcing bars. In addition, an air gap, with constant thickness, is included between the steel deck and the bottom surface of the concrete, to simulate the separation effect of both materials (debonding). Effectively, the thermal contact resistance in the joint boundary of layers is an important parameter that is not considered by most of authors [17]. This air gap model is introduced to simulate a similar thermal contact resistance.

The thermal models are validated with results from experimental tests found in the literature [7,8]. The thermal analysis on the steel deck components and on the reinforcing bars is evaluated in every composite slab with different concrete thicknesses $h_1$. The numerical results are compared with the simplified method from Annex D of Eurocode 1994-1-2. Finally, the results of all simulations developed for the parametric study are used to propose a new formula for the analytical estimation of the temperature of the four slab components. The coefficients of the new proposal are obtained by fitting this proposal to the numerical data using a non-linear least-squares method. It is important to mention that the proposed analytical method avoids unsafe temperature predictions resulting from the use of the simplified method, currently proposed by the standards.

This article is structured as follows. Section 2 introduces the thermal problem to be solved. The methodology used to solve it numerically is presented in Section 3. Section 4 is devoted to the simplified method provided by the standard for calculating the temperatures in the composite slab components. A new proposal for this simplified method, based on the numerical results obtained with the finite element calculation model, is presented in Section 5. This section also includes the validation of the model with experimental results available in the literature and a brief analysis of its computational performance. The article ends with the presentation of some final considerations in Section 6. In order to increase the readability of this paper, nomenclature has been defined in Table 1.

**Table 1.** Nomenclature.

| Composite Slab Geometry | |
|---|---|
| $l_1, l_2, l_3$ | Specific dimensions of the trapezoidal or re-entrant steel deck profile (mm) |
| $A$ | Concrete volume of the rib per metre of rib length ($mm^3/m$) |
| $L_r$ | Exposed area of the rib per metre of rib length ($mm^2/m$) |
| $u_1, u_2, u_3$ | Distances from the centre of the rebar to three points of the steel-deck (mm) |
| $h_1$ | Height of the concrete part of a composite slab above the decking (mm) |
| $h_2$ | Height of the concrete part of a composite slab within the decking (mm) |
| $\alpha$ | Angle of the web (°) |
| $\phi, \phi_{up}, \phi_{web}$ | View factors of the lower and upper flange and web |
| **Thermal Parameters** | |
| $T, T_\infty, T_{ISO}$ | Temperature, gas temperature, and standard ISO-834 fire temperature (°C ) |
| $\mathbf{T}, \mathbf{T}^0$ | Temperature vector and initial temperature vector |
| $\dot{\mathbf{T}}$ | Vector of the temperature derivatives |
| $\rho(T)$ | Specific mass ($kg/m^3$) |
| $C_p(T)$ | Specific heat (J/kgK) |
| $\lambda(T)$ | Conductivity (W/mK) |
| $\alpha_c$ | Convection coefficient |
| $\epsilon_m, \epsilon_f$ | Emissivity of the material and of the fire |
| $\sigma$ | Stefan–Boltzmann constant ($5.67 \times 10^{-8}$ ($W/m^2K^4$)) |
| $\theta_s$ | Temperature on the steel deck components provided by the standard EN1994-1-2 |
| $b_0, b_1, \ldots b_4$ | Coefficients provided by the standard EN1994-1-2 to estimate $\theta_s$ |
| $\theta_r$ | Temperature on the rebar provided by the standard EN1994-1-2 |
| $c_0, c_1, \ldots c_5$ | Coefficients provided by the standard EN1994-1-2 to estimate $\theta_r$ |
| $\theta_{sn}$ | New proposal to estimate the temperature on the steel deck components |
| $b_0, b_1, \ldots b_5$ | Coefficients for the new proposal of $\theta_{sn}$ |
| $\theta_{rn}$ | New proposal to estimate the temperature on the rebar |
| $c_0, c_1, \ldots c_6$ | Coefficients for the new proposal of $\theta_{sr}$ |
| $n$ | Number of temperature observations along time |
| $x_i, i = 1, \ldots n$ | Experimental values of the temperature in a given point of the slab |
| $y_i, i = 1, \ldots n$ | Numerical values of the temperature in a given point of the slab |
| **Mathematical Symbols and Operators** | |
| $t, t_0, t_f$ | Time, initial time, and final time (s) |
| $\nabla = (\partial_x, \partial_y, \partial_z)$ | Gradient |
| $\overrightarrow{n}$ | Unitary normal vector |
| $\mathbf{C}$ | Capacitance matrix |
| $\mathbf{K}$ | Conductivity matrix |
| $\mathbf{F}$ | Thermal load vector |
| **Acronyms and Abbreviations** | |
| E | Integrity criterion of slab resistance to fire |
| I | Insulation criterion of slab resistance to fire |
| R | Load-bearing criterion of slab resistance to fire |
| LWC | Lightweight concrete |
| NWC | Normal weight concrete |
| FEM | Finite elements method |
| Bias | Bias error |
| RMSE | Root-mean-squared error |
| SDE | Standard deviation of the error |

## 2. Heat Transfer Problem

This section is devoted to the development of the computational thermal model used to solve the non-linear transient thermal problems. The energy equation is solved in the multi-domain region corresponding to the composite slab under standard fire conditions. It is worth noting that the heat flux applied to the unexposed surface depends on the room temperature, and the heat flux applied to the exposed side to fire depends on the fire curve established by the ISO-834 fire standard [3].

## 2.1. Physical Multiple Domains

The three-dimensional (3D) transient heat transfer problems are solved on four composite slabs with different steel deck geometries, which are depicted in Figure 2. Two composite slabs with a trapezoidal geometry were selected (Confraplus 60 and Polydeck 59 s) and the other two slabs with re-entrant geometry (Multideck 50 and Bondek), with different values of the concrete topping thickness $h_1$.

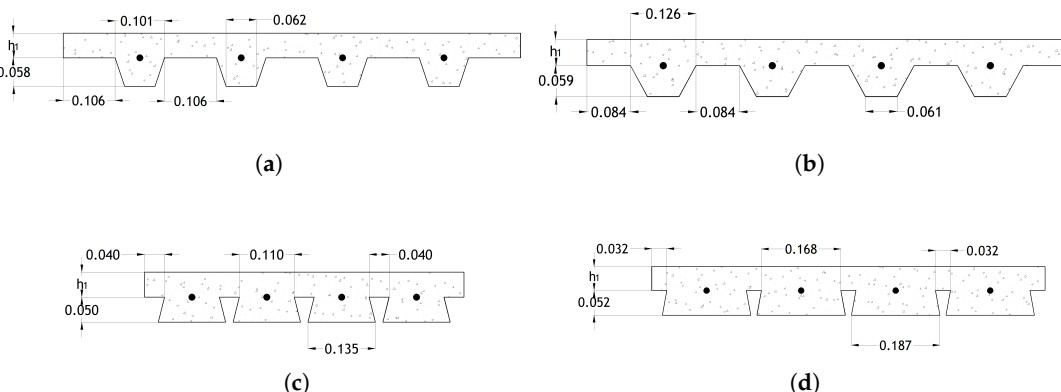

**Figure 2.** Composite slabs' geometric characteristics and dimensions (m). (**a**) Confraplus 60. (**b**) Polydeck 59S. (**c**) Multideck 50. (**d**) Bondek.

The 3D computational models were developed as close as possible with a realistic representation of the composite slabs. The geometry of the model considers the exact shape of the surfaces from a representative volume of the slab. The cross-section was selected by the side edges, delimited by the centre of the upper flange, comprising one rib, and part of the anti-crack mesh. The length of the specimens is 200 mm and does not affect the results. The geometries of the four representatives volumes modelled in Matlab are presented in Figure 3.

The multi-domain developed comprises five sub-domains: the steel deck, the air gap, the concrete, the rebars, and the anti-crack mesh. Thus, the materials that compose the physical sub-domains of the slabs are carbon steel (steel deck, rebars, and anti-crack mesh), concrete, and air (air gap to simulate the debonding effect between the steel deck and concrete). In addition, two types of concrete are used in composite slabs: NWC and LWC.

Confraplus 60 is a trapezoidal steel deck profile produced by ArcelorMittal. The steel deck is made with S350 steel grade, and the model uses a 1.25 mm thickness. The geometry is depicted in Figure 2a. The other ArcelorMittal composite slab is the Polydeck 59S model. This second model from ArcelorMittal is depicted in Figure 2b and uses a steel deck with S450 steel grade and a 1 mm thickness. The third geometry is a re-entrant model, presented in Figure 2c, based on the Multideck 50, produced by Kingspan Structural Products. This model has a steel deck with steel grade S450 and a 1 mm thickness. The second type of re-entrant composite slab under analysis is Bondek. This model is developed by the Lysaght company; the steel deck uses a steel grade S350 and considers a 1 mm thickness (see Figure 2d). These models were selected based on the current use and geometric differences.

The heat conduction inside this physical domain is mathematically modelled by the energy conservation equation:

$$\rho(T)C_p(T)\frac{\partial T}{\partial t} = \nabla \cdot (\lambda(T)\nabla T), \tag{1}$$

where $T$ represents the temperature (°C), $\rho(T)$ is the specific mass (kg/m$^3$), $Cp(T)$ is the specific heat (J/kgK), $\lambda(T)$ is the thermal conductivity (W/mK), $t$ is the time (s), and $\nabla = (\partial_x, \partial_y, \partial_z)$ is the gradient. Equation (1) is based on the heat flow balance, for the infinitesimal material volume, in each spatial direction.

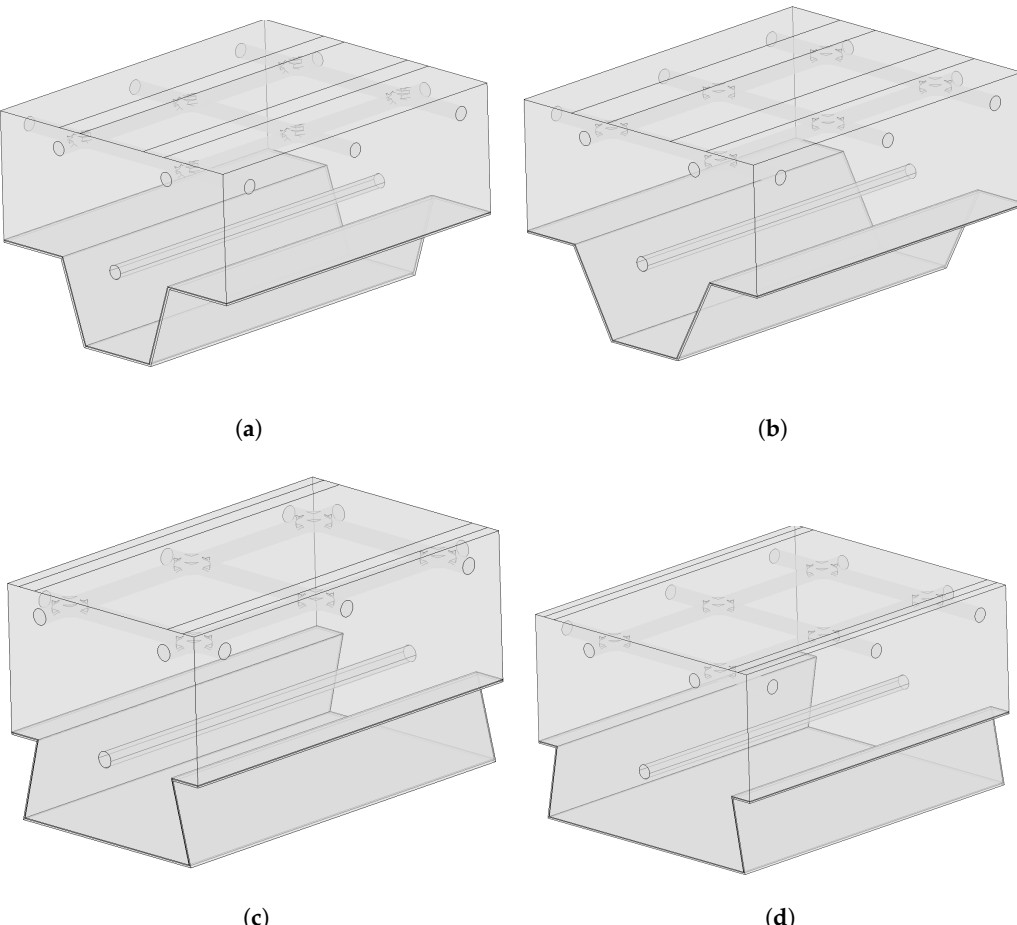

**Figure 3.** Geometry of the representative volumes modelled in Matlab. (**a**) Confraplus 60. (**b**) Poly-deck 59S. (**c**) Multideck 50. (**d**) Bondek.

The thermal properties ($\rho(T)$, $Cp(T)$, and $\lambda(T)$) of the materials that compose the slabs are determined by the Eurocodes [4,18,19] (steel and concrete) and by Cengal and Ghajar [20] (air) and are temperature dependent. Therefore, the specific mass $\rho(T)$, the specific heat $Cp(T)$, and the thermal conductivity $\lambda(T)$ vary with the temperature, introducing the non-linearity of Equation (1).

Although some studies consider that the thermal properties are constant, they are highly temperature dependent (see, for example, [21], for the case of thermal conductivity). Figure 4 describes the variation of the thermal properties, for the four different types of materials that constitute the studied composite slabs.

As the material properties depend on the temperature, there are important changes in the shapes presented in Figure 4, which are related to the material transformation. For steel, the most important shape modification is related to the transformation of the phase $\alpha$ (ferrite) to phase $\gamma$ (austenite). This effect is responsible for the modification of the specific heat ($C_p$) and conductivity $\lambda$, around 600–800 °C (Figure 4c). In the case of NWC, there are several physical events, but the most important are related to the moisture evaporation (free and bonded). This effect changes the material properties around 100 °C (Figure 4a).

Considering that the heat flux, received by the fire-exposed surface, changes with time, Equation (1) is time dependent and holds a non-linear transient thermal state for the analysis. Therefore, in order to determine the temperature field along time, the solution of Equation (1) is required. Furthermore, for the correct solution under analysis, the boundary conditions are applied according to the bulk temperature evolution from the ISO-834 standard fire curve [22].

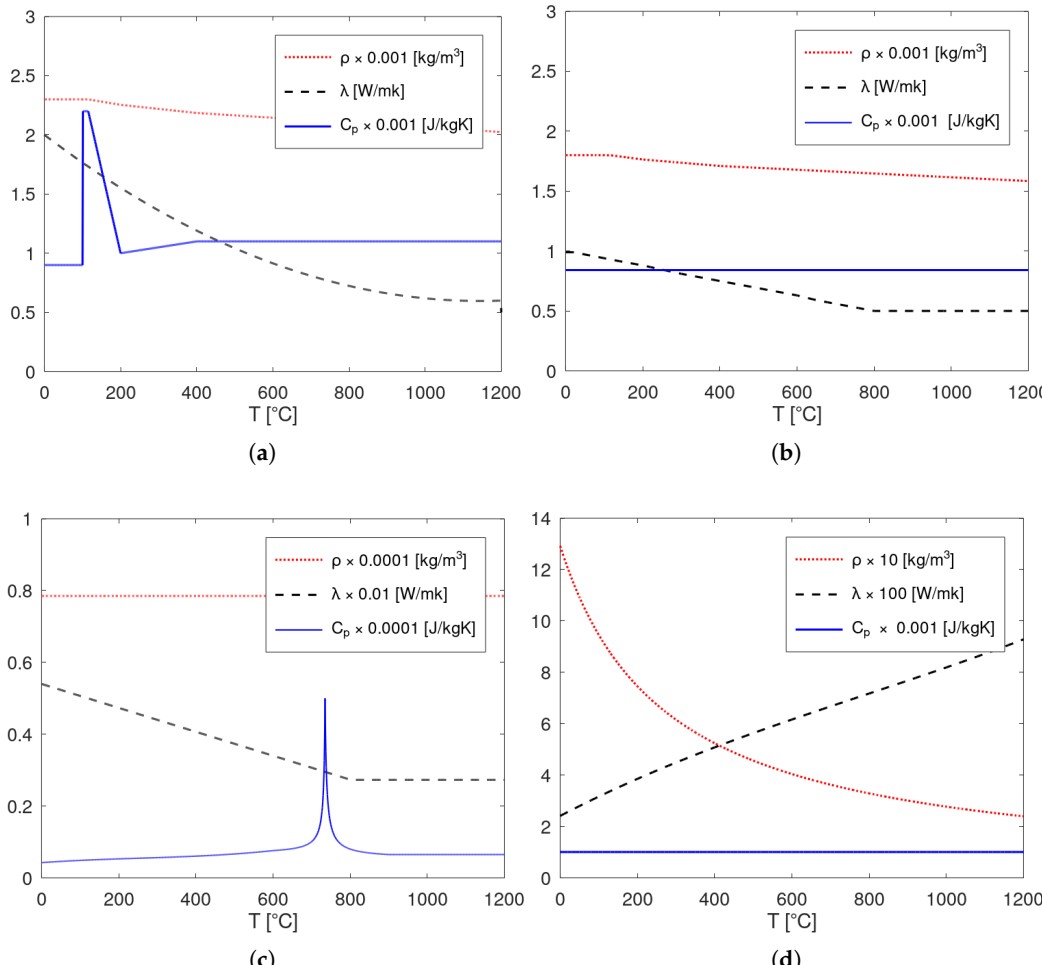

**Figure 4.** Thermal properties. (**a**) NWC. (**b**) LWC. (**c**) Carbon steel. (**d**) Air.

### 2.2. Boundary Conditions

To define the boundary conditions, it is necessary to understand the different modes of heat transfer that affect the composite slabs, that is the heat flux by conduction, convection, and radiation. In thermal analysis, the finite element mesh is generally used to model solids in which conduction is the predominant heat transfer mode, while the heat flux modes by radiation and convection are imposed through boundary conditions. The composite slabs are subjected to three main boundary conditions comprising the exposed surface, the unexposed surface, and the insulated surface. The boundary conditions use the parameters defined in Eurocode EN1991-1-2 [4].

The boundary conditions in the exposed side of the slab comprise the heat flux by convection and radiation and are given by

$$\lambda(T)\nabla T.\overrightarrow{n} = \alpha_c(T_\infty - T) + \phi\epsilon_m\epsilon_f\sigma\left(T_\infty^4 - T^4\right) \tag{2}$$

where $\overrightarrow{n}$ is the unitary vector normal to the external surface, $\phi$ is the view factor, $\alpha_c$ is the convection coefficient, $\epsilon_m$ is the emissivity of the material, $\epsilon_f$ is the emissivity of the flames, $\sigma$ is the Stefan–Boltzmann constant, and $T_\infty$ is the gas temperature of the fire compartment (bulk temperature).

In Equation (2), the convection coefficient is $\alpha_c = 25$ W/m$^2$K, the emissivity of steel is $\epsilon_m = 0.7$, and the fire emissivity is $\epsilon_f = 1$. This equation represents the amount of energy (or the heat flux) that arrives at the steel deck by radiation and convection based on the gas

bulk temperature, which will be transferred through the slab by conduction. The boundary conditions' parameters are represented in Figure 5.

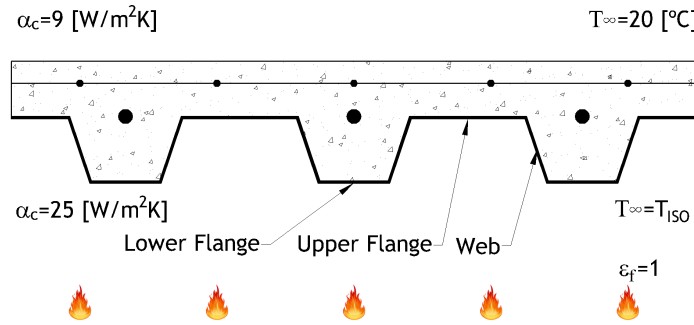

**Figure 5.** Boundary conditions.

The view factor ($\phi$) is a term of great relevance in studying the thermal behaviour of structures exposed to fire, which quantifies the geometric relation between the surface-emitting radiation and the receiving surface. Thus, it is a non-dimensional parameter and depends on the rib surface orientations and the distance between the radiative surfaces. The view factor for the lower flange is 1, and the values for the web and upper flange are usually smaller than 1. The view factor changes from point to point, over the exposed surface, due to the complexity of the geometry. The view factor associated with the web and upper flange components of steel deck is approximated by the crossed-strings method, proposed by Hotell H. C. in 1950 [20]. Figure 6 shows the parameters required for the calculation of the view factor in composite slabs.

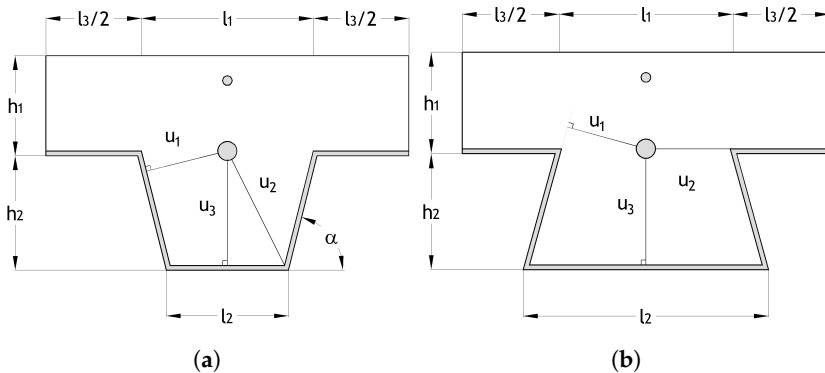

**Figure 6.** Geometric parameters used to determine view factors according to each slab profile. (**a**) Trapezoidal. (**b**) Re-entrant.

The view factor of the lower flange is $\phi = 1$, and the resulting equations for the upper flange ($\phi_{\text{up}}$) and web ($\phi_{\text{web}}$) view factors are presented in Equations (3) and (4).

$$\phi_{\text{up}} = \frac{\sqrt{h_2^2 + \left(l_3 + \frac{l_1 - l_2}{2}\right)^2} - \sqrt{h_2^2 + \left(\frac{l_1 - l_2}{2}\right)^2}}{l_3} \tag{3}$$

$$\phi_{\text{web}} = \frac{\sqrt{h_2^2 + \left(\frac{l_1 - l_2}{2}\right)^2} + (l_3 + l_1 - l_2) - \sqrt{h_2^2 + \left(l_3 + \frac{l_1 - l_2}{2}\right)^2}}{2\sqrt{h_2^2 + \left(\frac{l_1 - l_2}{2}\right)^2}} \tag{4}$$

Equation (5) presents the bulk temperature evolution of the fire compartment, which follows the standard fire curve ISO-834 ($T_\infty = T_{\text{ISO}}$), given by

$$T_{\text{ISO}} = 20 + 345 \log_{10}(8t + 1), \tag{5}$$

where $T_{\text{ISO}}$ is given in °C and $t$ in min [3].

The unexposed surface of the composite slab is also an important side to determine the temperature evolution. After all, it will determine the heat transfer from the compartment below the composite slab to the above compartment. Following the recommendations of Eurocode EN1991-1-2, the boundary condition on the unexposed side may be defined by the heat flux by convection, using $\alpha_c = 9 \text{ W/m}^2\text{K}$, to include the radiation effect [18]. This boundary condition in the unexposed surface of the slab is given by Equation (6),

$$\lambda(T)\nabla T.\overrightarrow{n} = \alpha_c(T - T_\infty) \tag{6}$$

where $T_\infty$ is the room temperature.

The adiabatic boundary conditions represented by Equation (7) are applied to the other four surfaces of the slab (front, back, left, and right).

$$\lambda(T)\nabla T.\overrightarrow{n} = 0. \tag{7}$$

## 3. Numerical Solution through Finite Element Method

In thermal problems, the finite element mesh is generally used to model solids in which conduction is the predominant heat transfer method, and the radiation and convection are imposed through boundary conditions. To perform a thermal analysis on composite slabs, Equation (1) is discretised by finite elements inside the physical sub-domains, corresponding to different materials. Figure 7 presents the generated mesh, produced by the Matlab PDE Toolbox, for each composite slab.

Equation (1) is solved by the finite elements method (FEM), based on the weak-form Galerkin model, using the weighted residual method, thus leading to the energy matrix formulae:

$$\mathbf{C}(\mathbf{T})\dot{\mathbf{T}} + \mathbf{K}(\mathbf{T})\mathbf{T} = \mathbf{F} \tag{8}$$

where $\mathbf{C}$ is the capacitance matrix, $\dot{\mathbf{T}}$ is the vector of the time derivatives of the nodal temperatures, $\mathbf{K}$ is the conductivity matrix, and $\mathbf{F}$ is the vector of the thermal loads (for details, see, for instance, [23]). The vector of the thermal loads $\mathbf{F}$ includes the boundary conditions. The solution of the first-order non-linear system of ordinary differential equations given by Equation (8), considering $\mathbf{T}(t_0) = \mathbf{T}^0$ and the boundary conditions, enables determining the temperature at each node of the finite element mesh, illustrated in Figure 7, over the time interval $\left[t_0, \, t_f\right]$.

*Computational Solution with Matlab*

The Matlab (R2021a) PDE Toolbox was used to solve the non-linear transient thermal problem. The finite element model of the composite slab, created with Matlab, uses the tetrahedral finite element type. This finite element is defined by four nodes and uses linear interpolating functions. The finite element mesh includes the five sub-domains: concrete, steel deck, rebar, anti-crack mesh, and air gap. Thus, three different materials are used (concrete, steel, and air). Additionally, two different types of concrete are considered: NWC and LWC.

The minimum finite element mesh size length changes according to each composite slab profile. Figure 7 depicts the finite element mesh of the models, the element size being selected by a convergence test of the results, which means that the size of the mesh was adjusted until the relative error in nodal temperature calculations reached the maximum value of $10^{-3}$ in the worst scenario.

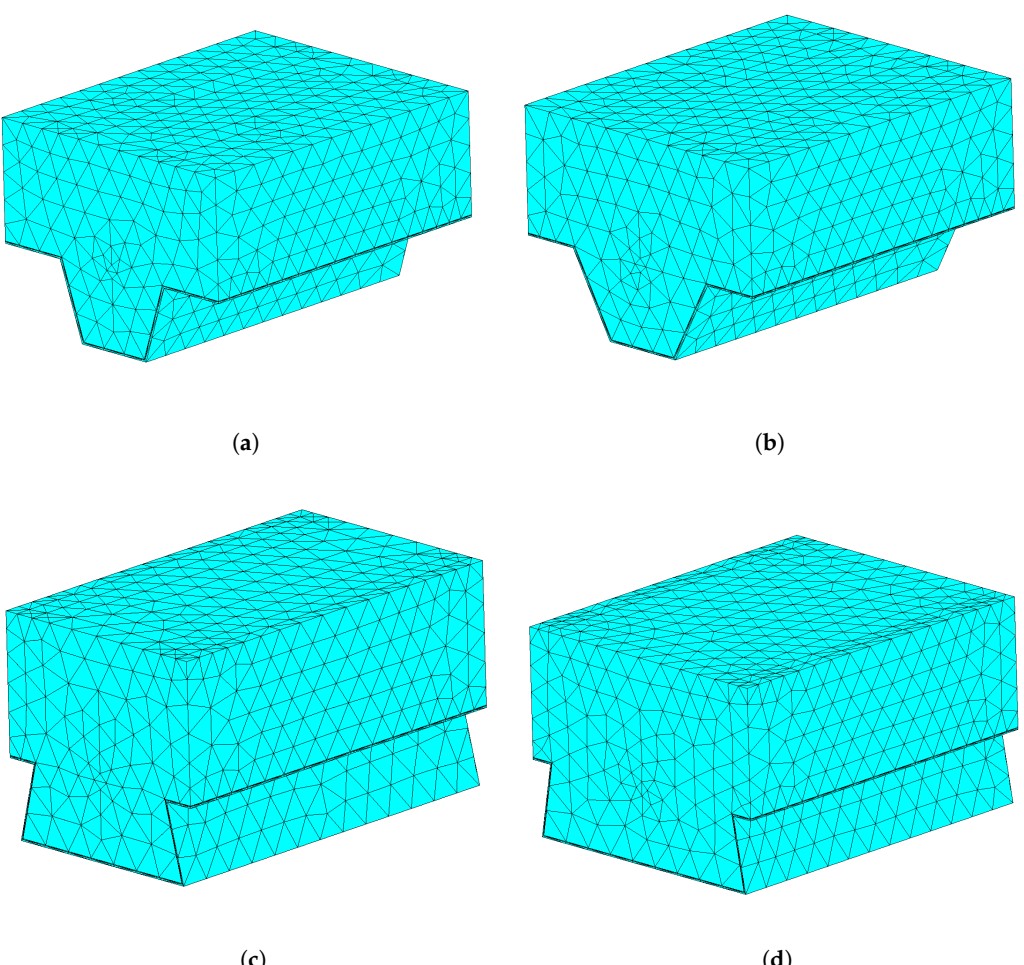

**Figure 7.** Finite elements mesh of the representative volumes modelled in Matlab. (**a**) Confraplus 60. (**b**) Polydeck 59S. (**c**) Multideck 50. (**d**) Bondek.

The solution method used by the Matlab PDE Toolbox is incremental in time and iterative in each time step due to the non-linear thermal properties of the material. Equation (8) is converted to

$$\mathbf{C}(\mathbf{T})\dot{\mathbf{T}} = \bar{\mathbf{F}}(\mathbf{T}) \tag{9}$$

where $\bar{\mathbf{F}}(\mathbf{T}) = \mathbf{F} - \mathbf{K}(\mathbf{T})$. The solution of Equation (9) is achieved by the built-in function `ode15s` [24]. The solution method implemented in this function is based on the discretisation of the time derivative by numerical differentiation formulas (NDFs) of orders 1 to 5. The accuracy of the solution can be explicitly controlled through the absolute or relative tolerance parameters. The absolute tolerance is a threshold parameter, below which the value of the solution component is disregarded. This property determines the accuracy when the solution approaches zero. The relative tolerance is a measure of the error relative to the size of each solution component (for more details, see [16]). In this paper, the absolute tolerance is set to $10^{-6}$ and the relative tolerance is set to $10^{-3}$. In each time step, the non-linear system of algebraic equations is solved through the Gauss–Newton method, whose stopping criterion can be monitored by means of the maximal number of iterations (25), the residual tolerance ($10^{-4}$), and the minimum damping of search direction ($10^{-3}$).

## 4. Simplified Calculation Method

The simplified calculation method used for the load-bearing criterion (R) presented in Eurocode EN1994-1-2 [4] can be applied to simply supported composite slabs when exposed to an ISO-834 standard fire [3].

In order to calculate the bending moment resistance of the composite slab (sagging moment), this standard requires the temperature calculation for each steel deck component (upper flange, web, and lower flange), for each required fire rating, according to the formula

$$\theta_s = b_0 + b_1 \frac{1}{l_3} + b_2 \frac{A}{L_r} + b_3 \phi + b_4 \phi^2 \tag{10}$$

and also for the rebar component $\theta_s$ by the formula

$$\theta_r = c_0 + c_1 \frac{u_3}{h_2} + c_2 z + c_3 \frac{A}{L_r} + c_4 \alpha + c_5 \frac{1}{l_3} \tag{11}$$

where the temperatures $\theta_a$ and $\theta_s$ are given in °C. The parameter $\phi$ is dimensionless and represents the view factor of the steel deck component (upper flange only), given by Equation (3); $l_3$ is the distance within the ribs (see Figure 6); $u_3$ represents the distance from the middle of the rebar to the lower flange in mm (see Figure 6); the $z$-factor represents the position of the rebar concerning the slab rib given by

$$\frac{1}{z} = \frac{1}{\sqrt{\frac{1}{u_1}}} + \frac{1}{\sqrt{\frac{1}{u_2}}} + \frac{1}{\sqrt{\frac{1}{u_3}}} \tag{12}$$

in mm$^{-0.5}$; $\alpha$ represents the angle between the web component of the steel deck and the horizontal direction in degrees (°) (see Figure 6); $\frac{A}{L_r}$ is the ratio between the concrete volume and the exposed area per meter of rib length of the steel deck, given in mm, and its calculation is performed through

$$\frac{A}{L_r} = \frac{h_2 \left( \frac{l_1 + l_2}{2} \right)}{l_2 + 2\sqrt{h_2^2 + \left( \frac{l_1 - l_2}{2} \right)^2}}. \tag{13}$$

The terms $b_i$ and $c_i$ represent the empirical coefficients given by Eurocode EN1994-1-2 [4], which depend on the type of concrete used (NWC or LWC) and on the standard fire rating, which must be verified.

Equations (12) and (13) enable estimating the temperatures that affect the reduction strength coefficients in all the slab components and, consequently, an accurate estimation of the load-bearing resistance criteria. The load-bearing resistance criteria (R) will be based on the reduction coefficients applied to the yield strength of each component.

## 5. Validation, Parametric Analysis, and New Proposal

After the validation of the numerical model, a parametric study was conducted to analyse the influence of the concrete thickness $h_1$ on the temperature of each component, used to determine the fire resistance of composite slabs according to the load-bearing criterion (R). As mentioned before, the load-bearing resistance depends on the temperatures in the steel deck components and rebar. A simplified calculation method is proposed by Eurocode EN1994-1-2 [4]. Unfortunately, this model has not been modified for a long time and, according to the authors, needs to be improved. The proposed methodology is based on the numerical solution of the non-linear and incremental thermal analysis, presented in the previous section, for different thicknesses of the concrete topping ($h_1$), comparing the results with the simplified solution method.

### 5.1. Validation of the Computational Model

Measuring the errors of numerical against experimental results is of paramount importance in order to assess the performance of the computational model. As shown by Chai and Draxler [25], assessing the accuracy of the predicted values is best done using

various metrics. Three metrics are especially useful when trying to assess the performance of a computational model. The simplest of those metrics is the bias, given by

$$\text{Bias} = \frac{1}{n} \sum_{i=1}^{n} (x_i - y_i), \tag{14}$$

where $n$ is the number of values, $x_i$ is a numerical value, and $y_i$ is the corresponding experimental value. As its name suggests, bias measures the bias of the model, which is simply the average error compared to the experimental. However, it does not really show the behaviour of the error. It is useful to determine if the computational model makes predictions that are lower or higher than the reference value (experimental), but it is not enough in itself to really know how well the model performs. It only shows the systematic error of the model.

Thus, complementary to the bias, the root-mean-squared error (RMSE) is also used:

$$\text{RMSE} = \sqrt{\frac{1}{n} \sum_{i=1}^{n} (y_i - x_i)^2}. \tag{15}$$

The RMSE is useful because the squared terms give a higher weight to higher errors. Thus, the RMSE will be higher if the model makes predictions that are far from the reference, even if these erroneous predictions are few in number.

To complement the bias and RMSE metrics, the standard deviation of the error (SDE) may also be considered. The SDE simply corresponds to

$$\text{SDE} = \sqrt{\text{RMSE}^2 - \text{Bias}^2}. \tag{16}$$

Considering the bias as a basic indicator of the systematic error in a prediction, then the SDE is the equivalent indicator of the random error.

Unfortunately, there are no experimental results in the literature with the four slabs that are investigated in this work (see Figure 2). Therefore, the finite element computational model was validated with the experimental results published by Lim and Wade [26] and Piloto et al. [7]. The results published by Lim and Wade [26] correspond to test number 4, obtained on a composite slab with a clear span of 3160 mm wide by 4160 mm long, with normal-weight concrete with siliceous aggregates and the moisture content of 5.6% by weight; the initial bulk temperature was 13 °C. The results published by Piloto et al. [7] correspond to test number 1, obtained on a slab with a clear span of 985 mm wide by 916.8 mm long; normal-weight concrete was used, and the moisture content was approximately 3.0% by weight; the initial bulk temperature amounted to 20 °C.

The values of the validation errors obtained with different air gap values are presented in Table 2, for the temperatures obtained at three points of the slab: P1, P2, and P3 (see the left-upper corner of Figure 8). In the case of Lim and Wade [26], the points are located at the middle of the rib, respectively, at distances of 20, 70, and 130 mm from the top. In the case of Piloto et al. [7], P1 and P3 are located in the middle of the upper flange, at distances of 20 and 15 mm, respectively, from the top; P2 is located in the middle of the rib at a distance of 15 mm from the top.

From the validation with the experimental results of Lim and Wade [26] presented in Table 2, one can see, in the case of P1, that the global reduction of the errors is achieved with an air gap of 0.5 mm, while for points P2 and P3, this reduction is not achieved; on the contrary, the errors increase with the increase of the air gap size. Figure 8a depicts the evolution of the experimental and numerical temperatures in points P1, P2, and P3 along the first 120 min after the beginning of the fire, for the case of an air gap of 0.5 mm. Additionally, the bulk temperature corresponding to the standard fire ISO 834, given by Equation (5), and the furnace temperature, registered during the experiment, are also included.

**Table 2.** Errors of the validation with experimental results.

| Air Gap: | 0 mm | | | 0.5 mm | | | 1 mm | | |
|---|---|---|---|---|---|---|---|---|---|
| Errors: | Bias | RMSE | SDE | Bias | RMSE | SDE | Bias | RMSE | SDE |
| Points | | | | Experimental Results from Lim and Wade [26] | | | | | |
| P1 | −89.4 | 107.0 | 58.8 | 21.1 | 36.9 | 30.2 | −119.8 | 158.0 | 103.1 |
| P2 | 56.7 | 66.1 | 34.0 | 80.9 | 92.1 | 44.2 | 102.2 | 117.0 | 57.0 |
| P3 | 58.1 | 66.8 | 33.0 | 70.0 | 81.5 | 41.9 | 79.3 | 93.6 | 49.7 |
| | | | | Experimental Results from Piloto et al. [7] | | | | | |
| P1 | −192.9 | 222.9 | 111.6 | −128.9 | −21.6 | −101.2 | −78.9 | −4.4 | 60.7 |
| P2 | −48.3 | 70.3 | 61.7 | 153.1 | 45.3 | 125.8 | 97.1 | 22.9 | 78.5 |
| P3 | −152.2 | 183.3 | 102.1 | 82.5 | 39.8 | 74.8 | 56.6 | 22.4 | 49.7 |

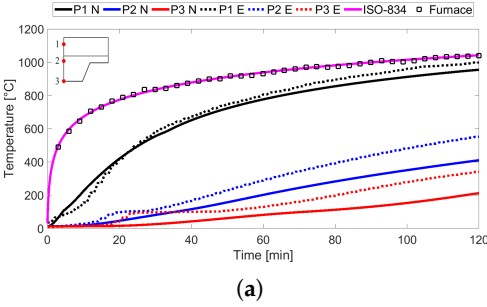
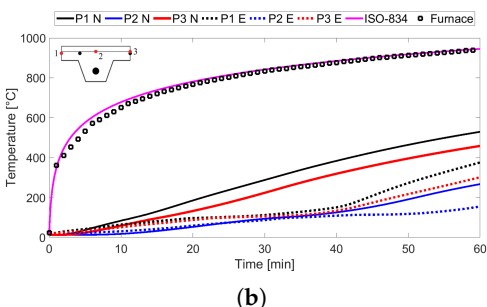

| (a) | (b) |

**Figure 8.** Temperature evolutions at points P1, P2, and P3 for the case of an air gap of 0.5 mm. (**a**) Experimental results from Lim and Wade [26]. (**b**) Experimental results from Piloto et al. [7]. E: experimental results, N: numerical results.

For the experimental results depicted in Figure 8, one can see a plateau at about 100 °C. This is due to moisture evaporation in the concrete. This phenomenon translates into a sudden rise in the specific heat (illustrated in Figure 4a) and has the immediate consequence of a decrease in the rate of temperature increase. The results of the numerical simulations do not present this pronounced plateau, probably because localised moisture concentrations in the tests were higher than the uniform moisture content introduced in the thermal models for each slab.

From the validation with the experimental results of Piloto et al. [7] presented in Table 2, it can be observed that, in general, the errors decrease with the increase of the air gap. Figure 8b depicts the experimental and numerical temperatures in points P1, P2, and P3 along the first 120 min after the beginning of the fire, for the case of an air gap of 0.5 mm, along with the bulk and furnace temperatures.

From the results of the validation presented in Table 2 and Figure 8, one cannot conclude that there exists an optimal air gap. The experimental results are highly influenced by the concrete thermal properties, such as the moisture content, which influences, in particular, the specific heat. For these reasons, an air gap of 0.5 mm, greater than zero, but not too thick, was chosen.

### 5.2. Parametric Analysis

A parametric analysis was developed to determine the influence of the concrete thickness $h_1$ on the temperature field, used to determine the fire resistance of composite slabs and the ability to sustain the load (R). A total of 40 numerical simulations were carried out, using an air gap of 0.5 mm, in agreement with the reasons previously discussed. The computational simulations took into account $h_1$ values equal to 60, 70, 90, 110, and 125 mm for the two trapezoidal steel deck geometries (see Figure 2a,b) and $h_1$ values equal

to 50, 70, 90, 110, and 125 mm for the two re-entrant steel deck geometries (see Figure 2c,d). All simulations were developed for NWC and LWC.

Figures 9–12 show the temperature field inside the composite slab after 120 min and the time evolution of the average temperatures, determined by numerical simulation (N), in the three components of the steel deck and in the rebar. For comparison, the values calculated using the simplified method (S), given by Equations (10) and (11), are also included. In the case of NWC, the standard EN-1994-1-2 provides only the fire rating for 60, 90, and 120 min, because it is considered that, if the slab project is well made for the room temperature, it is able to resist fire for at least 30 min. According to the same standard, when the slab is made of LWC, it will require the fire analysis for 30 min. Additionally, the $T_{\text{ISO}}$ given by Equation (5) is also depicted in these figures.

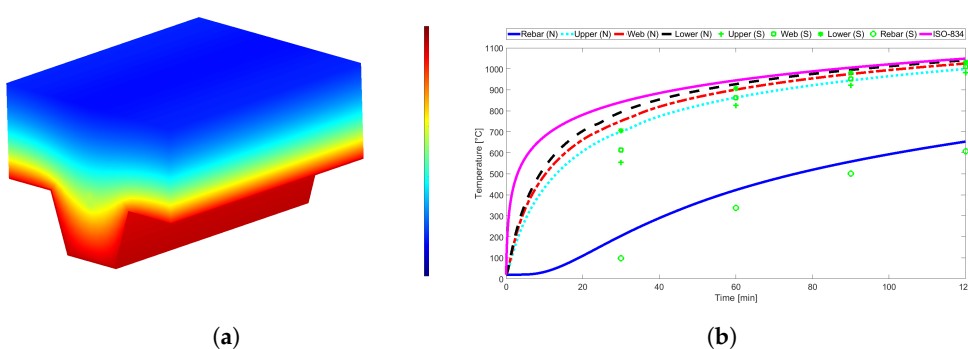

(**a**)                                            (**b**)

**Figure 9.** Confraplus 60 with LWC. (**a**) Final temperature distribution. (**b**) Temperature evolution: N stands for numerical and S for simplified method.

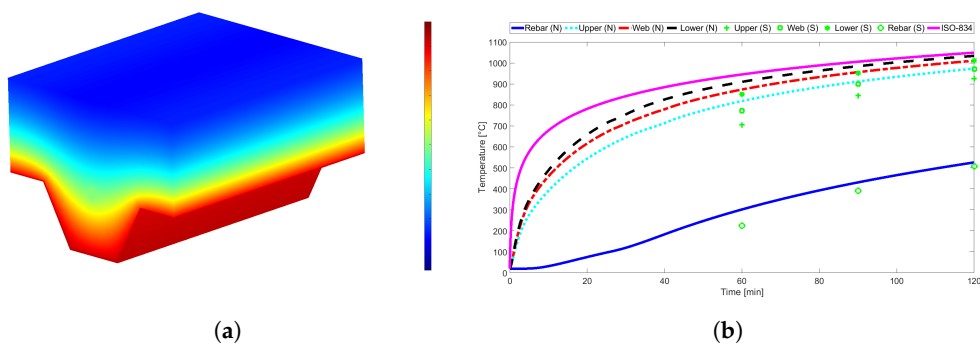

(**a**)                                            (**b**)

**Figure 10.** Polydeck 59S with NWC. (**a**) Final temperature distribution. (**b**) Temperature evolution: N stands for numerical and S for simplified method.

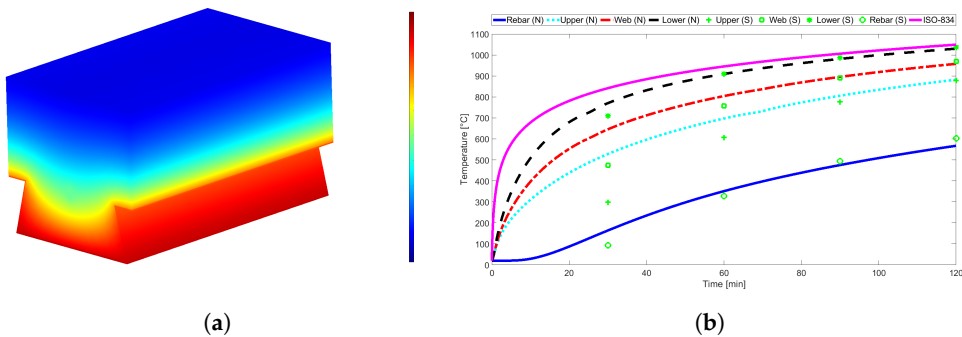

(**a**)                                            (**b**)

**Figure 11.** Multideck 50 with LWC. (**a**) Final temperature distribution. (**b**) Temperature evolution: N stands for numerical and S for simplified method.

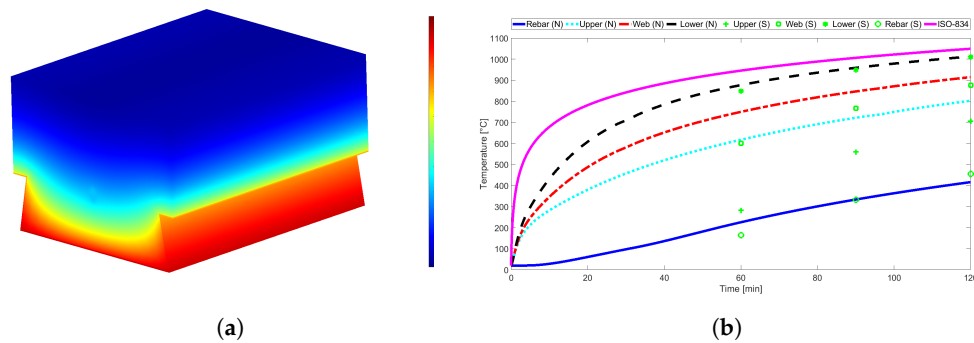

**Figure 12.** Bondeck with NWC. (**a**) Final temperature distribution. (**b**) Temperature evolution: N stands for numerical and S for simplified method.

Figure 9 shows the thermal behaviour of the trapezoidal Confraplus 60 slab with $h_1 = 90$ mm using LWC. Figure 10 shows the thermal behaviour of the trapezoidal Polydeck 59S slab with $h_1 = 90$ mm using NWC. Figure 11 presents the thermal behaviour of the re-entrant Multideck 50 slab with $h_1 = 90$ mm using LWC. Figure 12 presents the thermal behaviour of the re-entrant Bondeck slab with $h_1 = 90$ mm using NWC. One can see that, in general, the temperatures predicted by the simplified calculation method are below the numerical results. This is especially true for the fire ratings of 30 and 60 min, in the case of the Multideck 50, and for the fire rating of 60 min, in the case of the Polideck 59S geometry. It is also noted that the temperature in the rebar is higher when LWC is used instead of NWC. It can also be noted that the temperatures of the trapezoidal slabs are usually higher than the temperatures of the re-entrant slabs.

### 5.3. Computational Times

The computational simulations were performed on a Linux system, running in a machine with a 16-core AMD EPYC 7351 CPU and 64 GB of RAM, with a Matlab code purely sequentially. The computational times related to the simulation of the thermal effects, caused by a standard fire over 120 min, are shown in Table 3. These times were obtained in the four slabs with thickness $h_1 = 125$ mm. The anti-crack mesh is located 19 mm from the top. The space between each bar of the anti-crack mesh is 50 mm, and the diameter of each bar is 4 mm.

**Table 3.** Computational times (s) for concrete topping thickness $h_1 = 125$ mm.

| Slab Geometry | With Anti-Crack Mesh | | Without Anti-Crack Mesh | |
|---|---|---|---|---|
| | NWC | LWC | NWC | LWC |
| Confraplus 60 | 4044 s | 703 s | 3116 s | 479 s |
| Polydeck 59S | 4095 s | 722 s | 2823 s | 513 s |
| Multideck 50 | 4845 s | 838 s | 3747 s | 663 s |
| Bondek | 6682 s | 1086 s | 3414 s | 573 s |

The results presented in Table 3 show that the computational times of the simulations with NWC are much higher than the times obtained with LWC. This is due to the thermal properties of the LWC, which vary more linearly with temperature than the properties of NWC (see Figure 4), creating fewer convergence problems in the numerical integration scheme. It is also observed that the slabs with the re-entrant geometry present longer computation times. This is due to the fact that they present a larger volume, whose modelling implies the use of a greater number of finite elements.

Another important observation is that the use of the anti-crack mesh increases the computation times substantially. The reason is the refinement of the mesh and the consecutive increase in the number of nodes where it is necessary to estimate the temperature. However, this increase in time is not justified, as the estimation of temperatures in the slab

components is not improved. Figure 13 compares the numerical temperatures, obtained after 30, 45, 60, 90, and 120 min, with (A) and without (N) anti-crack mesh in the Confraplus 60 slab. The temperatures obtained with and without the anti-crack mesh are superposed, which means they are almost the same.

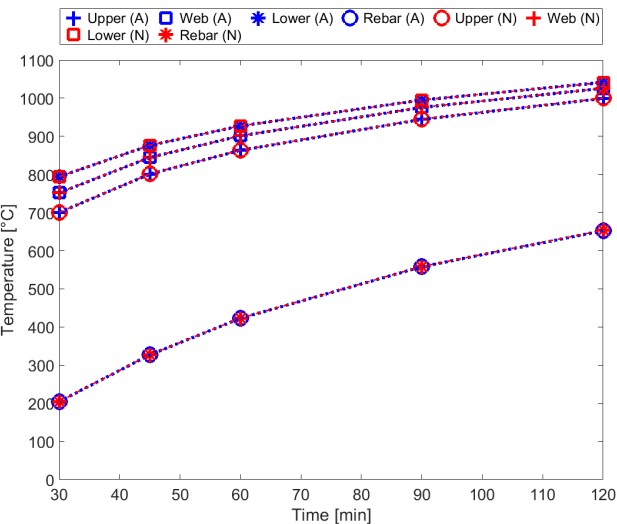

**Figure 13.** Numerical temperatures in the components of the Confraplus 60 slab, with (A) and without (N) anti-crack mesh.

*5.4. New Calculation Proposal*

Based on the numerical results, new coefficients $b_i$ and $c_i$ are proposed for the simplified calculation model. In addition, the original models, given by Equations (10) and (11), were modified by including a new term that depends on $h_1$, in order to take into account the effect on the temperatures for the thickness of the concrete topping. The thickness $h_1$ is explicitly included in the mathematical model multiplied by coefficient $b_5$, in the case of the steel deck temperatures, and by $c_6$, in the case of the rebar temperature. Thus, the new proposal for the steel deck temperature is

$$\theta_{sn} = b_0 + b_1 \frac{1}{l_3} + b_2 \frac{A}{L_r} + b_3 \, \phi + b_4 \, \phi^2 + b_5 \, h_1 \tag{17}$$

and for the rebar is

$$\theta_{rn} = c_0 + c_1 \frac{u_3}{h_2} + c_2 \, z + c_3 \frac{A}{L_r} + c_4 \, \alpha + c_5 \frac{1}{l_3} + c_6 \, h_1. \tag{18}$$

The coefficients for these new proposed methods were determined by fitting the mathematical models, represented by Equations (17) and (18), to the numerical results obtained with the parametric analysis when changing the $h_1$ dimension, applied on two different types of concrete and the four different composite slab geometries. The coefficients were determined using the non-linear least-squares method. This method consists of minimising the sum of the squared deviations between the temperatures obtained with Equation (17) or (18) and those obtained through numerical simulations. This sum was minimised using the generalised reduced gradient (GRG) non-linear solver [27].

The determined coefficients used for the calculation of temperatures at the steel deck components, through Equation (17), are presented in Table 4. For the calculation of the temperatures at the rebar, through Equation (18), the proposed coefficients are presented in Table 5. It is worth mentioning that in addition to the standard fire ratings of 60, 90, and 120 min for NWC and 30, 60, 90, and 120 min for LWC, the new proposal also comprises the coefficients for the fire rating of 45 min. This fire rating may also be regarded for the

fire classification of construction products and building elements, when using also the separating function.

**Table 4.** Coefficients of the new proposal for estimating the temperature of the steel deck components.

| Fire Rating | Flange | $b_0$ | $b_1$ | $b_2$ | $b_3$ | $b_4$ | $b_5$ |
|---|---|---|---|---|---|---|---|
| | | **Normal Weight Concrete—NWC** | | | | | |
| **45 min** | **Upper** | 139.97 | 620.75 | 7.70 | 1421.99 | −1204.07 | −0.06 |
| | **Web** | 404.16 | −1623.28 | 6.75 | 1109.91 | −1060.65 | −0.06 |
| | **Lower** | 860.85 | −2427.81 | 1.03 | −40.23 | 36.98 | 0.00 |
| **60 min** | **Upper** | 224.59 | −2852.60 | 10.85 | 1428.88 | −1312.53 | −0.10 |
| | **Web** | 599.88 | −13,427.37 | 17.20 | 327.35 | −522.00 | −0.03 |
| | **Lower** | 917.51 | −3173.16 | 2.05 | −47.72 | 15.26 | −0.01 |
| **90 min** | **Upper** | 578.71 | −18,369.16 | 22.96 | 139.66 | −322.28 | −0.18 |
| | **Web** | 542.03 | 633.99 | 4.62 | 1398.31 | −1361.20 | −0.07 |
| | **Lower** | 982.26 | −3077.46 | 2.37 | 3.33 | −52.43 | −0.02 |
| **120 min** | **Upper** | 691.06 | −14,595.54 | 18.27 | 212.08 | −340.01 | −0.27 |
| | **Web** | 666.10 | −416.23 | 4.84 | 1158.51 | −1146.72 | −0.11 |
| | **Lower** | 955.40 | 4109.18 | −3.87 | 492.99 | −407.78 | −0.02 |
| | | **Lightweight Concrete—LWC** | | | | | |
| **30 min** | **Upper** | 302.16 | −5233.64 | 10.99 | 614.94 | −529.27 | −0.01 |
| | **Web** | 491.68 | −3193.58 | 6.65 | 522.43 | −491.20 | 0.00 |
| | **Lower** | 871.28 | −4956.79 | 3.09 | −469.35 | 438.67 | 0.00 |
| **45 min** | **Upper** | 388.57 | −7171.21 | 12.85 | 755.49 | −743.72 | −0.04 |
| | **Web** | 486.20 | 1073.65 | 3.52 | 1137.25 | −1073.28 | −0.01 |
| | **Lower** | 868.15 | −1328.82 | 0.68 | 22.40 | −24.49 | 0.00 |
| **60 min** | **Upper** | 389.98 | 114.91 | 5.89 | 1325.82 | −1200.05 | −0.07 |
| | **Web** | 632.61 | −4880.46 | 8.49 | 662.60 | −722.17 | −0.02 |
| | **Lower** | 904.46 | −277.68 | −0.07 | 115.21 | −106.16 | −0.00 |
| **90 min** | **Upper** | 578.38 | −425.05 | 4.87 | 1042.21 | −944.71 | −0.12 |
| | **Web** | 629.76 | 5806.40 | −1.51 | 1364.79 | −1243.52 | −0.05 |
| | **Lower** | 986.62 | −834.24 | 0.45 | 39.83 | −44.26 | 0.00 |
| **120 min** | **Upper** | 686.56 | 1931.24 | 1.59 | 994.61 | −861.46 | −0.14 |
| | **Web** | 820.51 | −1457.40 | 4.08 | 632.94 | −642.81 | −0.06 |
| | **Lower** | 1043.00 | −1215.34 | 0.80 | −15.00 | 2.37 | −0.01 |

**Table 5.** Coefficients of the new proposal for estimating the rebar temperature.

| Fire Rating | $c_0$ | $c_1$ | $c_2$ | $c_3$ | $c_4$ | $c_5$ | $c_6$ |
|---|---|---|---|---|---|---|---|
| | **Normal Weight Concrete—NWC** | | | | | | |
| **45 min** | 99.82 | 100.20 | 106.00 | −11.83 | 2.07 | −3983.08 | −0.06 |
| **60 min** | −880.00 | 923.77 | 389.18 | −30.70 | 2.96 | −5263.73 | −0.12 |
| **90 min** | 117.69 | 961.63 | −526.70 | 28.09 | 0.74 | −5803.21 | −0.35 |
| **120 min** | −151.22 | 834.65 | 31.95 | −8.06 | 2.21 | −7000.32 | −0.60 |
| | **Lightweight Concrete—LWC** | | | | | | |
| **30 min** | −496.77 | 430.07 | 326.41 | −25.82 | 2.46 | −3419.70 | −0.01 |
| **45 min** | −2463.48 | 2829.01 | 49.42 | −9.87 | 2.02 | −4509.88 | −0.06 |
| **60 min** | 317.37 | 53.53 | 179.58 | −19.07 | 2.43 | −5491.27 | −0.13 |
| **90 min** | 528.37 | −181.30 | 395.25 | −33.50 | 2.90 | −6393.06 | −0.31 |
| **120 min** | −373.73 | −325.98 | 1616.52 | −112.23 | 6.05 | −7756.37 | −0.44 |

Analysing the new coefficients presented in Tables 4 and 5, it appears that increasing $h_1$ leads to small reductions in temperatures on the steel deck components and on the rebar since $b_6$ and $c_6$ are generally small in magnitude. However, in the case of rebar, $h_1$ has a greater effect, especially for longer fire resistance ratings.

### 5.5. Comparison of Results

This subsection presents the differences between the results proposed by the simplified model (S) presented in Eurocode 1994-1-2 [4], the numeric results (N), and the new calculation proposal (P). This comparison is presented in the following graphs. Figure 14a establishes this comparison for each steel deck component and the rebar for LWC, while Figure 14b plays the same role for NWC, where both slabs have a trapezoidal profile. Figure 15 establishes the same comparison in the case of the two slabs with re-entrant profiles.

The results depicted in Figures 14 and 15 show that the new proposal fits very well with the proposed numerical results. It was observed also that the temperatures estimated by the simplified model are always lower than the numerical temperatures. The differences are most noticed for the lower fire ratings, especially in the case of steel deck components. The numerical temperatures on the rebar are relatively close to the temperatures estimated by the simplified model, especially for the slabs with the re-entrant geometry.

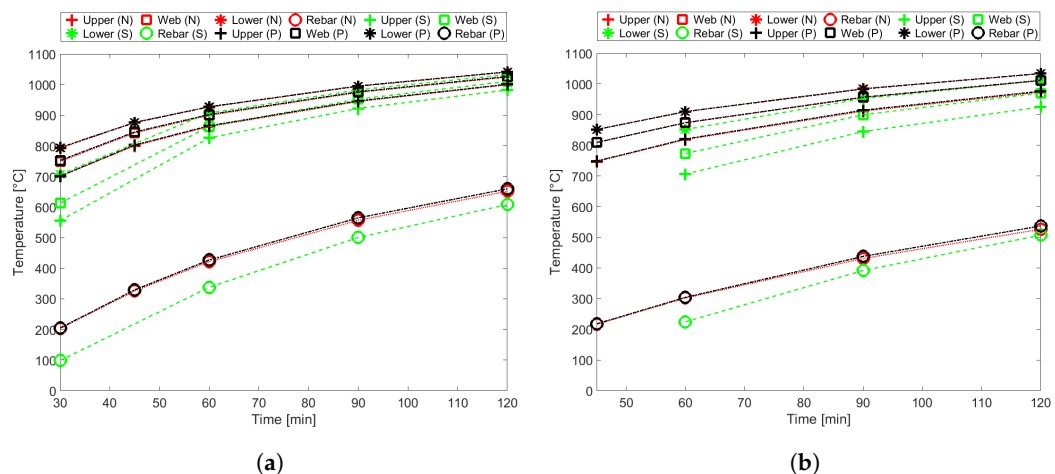

**Figure 14.** Temperature corresponding to different fire rating times. (**a**) Confraplus 60 with LWC. (**b**) Polydeck 59S with NWC. N stands for numerical, S for simplified method, and P for new proposal.

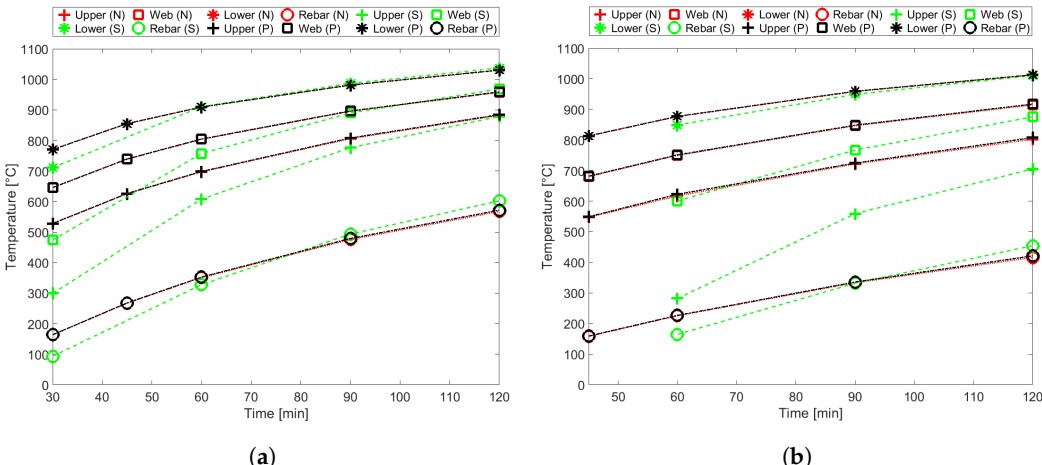

**Figure 15.** Temperature corresponding to different fire rating times. (**a**) Multideck 50 with LWC. (**b**) Bondeck 50 with NWC. N stands for numerical, S for simplified method, and P for new proposal.

When compared with the actual version of the Eurocode 1994-1-2 model, the new proposal contributes to improving the temperature estimation according to each fire rating time. For both types of concrete (NWC and LWC), the new proposal matches very well with the numerical results. It is also possible to verify that the simplified model, proposed by the current version of the Eurocode, predicted much lower temperatures than the numerical results, mainly for the case of steel deck components and for the lowest fire ratings. The new proposal decreases the difference between the expected temperatures and the numerical simulation results, by more than 100 °C, especially for low fire ratings. In the case of the rebar, the difference between the temperatures is smaller, especially for the slabs with the re-entrant geometry.

## 6. Final Considerations

In this paper, a realistic computational model, based on 3D finite elements, was developed to determine the thermal behaviour of composite slabs under standard fire conditions. This model takes into account the various sub-domains of the composite slab, their respective thermal properties, and the debonding effect of the steel deck. The results of the numerical simulations enabled us to determine the temperatures for different slab geometries and compare them with the ones provided by the simplified calculation model presented in Eurocode EN1994-1-2. The proposed model aims to include some effects that the current version of the Eurocode neglects, such as the debonding effect between the steel deck and concrete and the thickness of the concrete topping.

The computational model is reliable and allows thermal simulations on composite slabs to be carried out in a not very long time. However, these computational times can be reduced, without loss of the quality of the results, by simplifying the physical model. It thus constitutes an efficient alternative to carrying out experimental tests, which are very demanding and expensive.

In order to improve the current simplified method, new analytical formulae were proposed to determine the temperature in the steel deck components and in the rebars, based on the numerical results. The new proposal includes, with respect to the current model of the Eurocode, an additional term to account for the effect of the concrete thickness ($h_1$). The new coefficients were determined by fitting the new proposal to the numerical results by non-linear least squares.

The new proposal provides an improvement of the temperature estimation that affects the reduction coefficients in all the slab components. In the cases of low fire ratings, the new formulae decrease the difference between the expected temperatures and the numerical simulation results, by more than 100 °C. The use of this new proposal enables accurate estimation of the load-bearing resistance criteria and, consequently, the safer design of composite slabs.

**Author Contributions:** Conceptualisation, C.B. and P.A.G.P.; methodology, C.B. and P.A.G.P.; software, C.B. and V.M.; validation, P.A.G.P., M.S., and V.M.; formal analysis, P.A.G.P. and C.B.; investigation, C.B., M.S., and V.M.; resources, C.B. and P.A.G.P.; data curation, P.A.G.P.; writing—original draft preparation, C.B.; writing—review and editing, C.B. and P.A.G.P.; visualisation, M.S. and V.M.; supervision, C.B. and P.A.G.P.; project administration, C.B. and P.A.G.P.; funding acquisition, C.B. and P.A.G.P. All authors have read and agreed to the published version of the manuscript.

**Funding:** This research received no external funding.

**Institutional Review Board Statement:** Not applicable.

**Informed Consent Statement:** Not applicable.

**Data Availability Statement:** Not applicable.

**Conflicts of Interest:** The authors declare no conflict of interest.

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
