# Peer review of "Modelling the Thermal Effects on Structural Components of Composite Slabs under Fire Conditions"

_computation, doi:10.3390/computation10060094_

Round 1

Reviewer 1 Report

Reviewer Comment for Author

Journal: Computation

Manuscript ID: computation-1678764

Title: Modelling the Thermal Effects on Structural Components of Composite

Recommendation: Major revision

The manuscript presents interesting achievements in the field. It may be recommended to be published in the Computation Journal. However, the paper is not well written and the paper has many weaknesses. The idea and the results need more depth.

  1. The novelty of the work should be highlighted to real physics phenomena in both the introduction and abstract. This kind of problem has been studied many times and in the same way.
  2. Model the heat conduction within this physical field mathematically by the energy conservation equation 1. Is Equation 1 called the Energy Conservation Equation? Also, this equation is of parabolic type and therefore it can predict infinite speeds of heat. Please explain.
  3. The mathematical model is unclear as well as the solution method.
  4. The discussion is sketchy. It is a more graphical presentation of various numerical computations, and lacks depth and physical contents. The most important results obtained should be clearly highlighted.
  5. Physical and thermal constants such as specific heat, density, and thermal conductivity coefficient are variable functions that depend on heat, but the shape of the change function is not clear. Please explain.
  6. Extensive editing of English language and style is required. Numerous grammatical and spelling errors greatly degraded the quality of the presentation.
  7. There exist some un-defined acronyms through the text, which need to be properly declared.
  8. In the discussion section, why was there no comparison between the different models, especially since the model used has flaws like the classic model of heat transport equation? You can use DOI: 10.1177/1464420720985899 , https://doi.org/10.1002/zamm.202000344
  9. Write the conclusion more precious.

Reviewer 2 Report

Title: Modelling the Thermal Effects on Structural Components of Composite Slabs Under Fire Conditions

MS NO: computation-1678764

In this study, a realistic computational model, based on 3D finite elements, was developed to determine the thermal behavior of composite slabs under standard fire conditions. The following comments should be addressed before a final decision.

  The abstract is too long. This section should be written more concisely.

·         What is the reason for jumping Cp diagrams in Fig. 4a and Fig. 4c?

·         Why is the finite element method used for numerical simulation?

·         Nomenclature should be added in order to define all variables and abbreviations used in the paper.

·         It is suggested to argue about heat transfer in composite material in more detail to enrich the literature review. The following review paper is suggested to refer to in this regard: A Comprehensive Review on Multi-Dimensional Heat Conduction of Multi-Layer and Composite Structures: Analytical Solutions, Journal of Thermal Science 30,1875-1907, 2021. [10.1007/s11630-021-1517-1].

·         The article argues about debonding composites. It is recommended to add some papers related to functionally graded material and introduce them to readers as a new version of composite structures without “debonding”. The following papers will be useful to introduce heat transfer in FGM: DOI: 10.1016/j.ijheatmasstransfer.2019.118515; 10.1016/j.icheatmasstransfer.2019.104280

Round 2

Reviewer 1 Report

After reviewing the revised version, it was found that the authors made the required improvements. They also responded to many concerns.